# Changes in Physical Activity and Depression among Korean Adolescents Due to COVID-19: Using Data from the 17th (2021) Korea Youth Risk Behavior Survey

**DOI:** 10.3390/healthcare11040517

**Published:** 2023-02-09

**Authors:** Yong-Sook Eo, Myo-Sung Kim

**Affiliations:** 1College of Nursing, Dongguk University, Gyeongju 38066, Republic of Korea; 2Department of Nursing, Dong-Eui University, Busan 47340, Republic of Korea

**Keywords:** adolescents, COVID-19, health behavior, mental health, physical activity

## Abstract

This study aimed to identify changes in the health behavior and mental health of adolescents due to coronavirus disease 2019 (COVID-19) and the characteristics related to changes in physical activity and depression among health behavior changes. Data were extracted from the 17th Korea Youth Risk Behavior Web-based Survey of 54,835 adolescents. We classified the adolescents into three groups according to changes in physical activity and depression: no change, increased, or decreased. Independent variables included changes in health behavior due to COVID-19, demographic characteristics, health-related behavior, and mental health. Data were analyzed using the χ2-test and multiple logistic regressions using the SPSS Statistics 27 program. Changes in physical activity and depression showing negative changes due to the pandemic were related to factors such as having breakfast, current smoking, current drinking, stress, loneliness, despair, suicidal ideation, suicide plans, and suicide attempts. The related factors differed between the increased and decreased groups. The results of this study confirm that it is necessary to develop programs to promote the health of youth by considering the factors that affect physical activity and depression, which in turn influences the status of their health.

## 1. Introduction

Since its first outbreak in 2019, coronavirus disease 2019 (COVID-19) emerged as a global pandemic and it continues to this day owing to the occurrence of new mutations [1]. Due to the severity of the pandemic, Korea implemented social distancing rules—the opening of schools was postponed for health and safety reasons, daily school activities were suspended due to remote learning, and overall time spent at home increased. As a result of these adjustments, teenagers experienced physical and psychological changes in daily life [2,3]. It has been shown that COVID-19 affects the school lives of adolescents as well as their psychological, emotional, health, and physical development [4]. In addition, these effects show differential effects on gender, age, region, and family background [4,5].

Several studies around the world also have found that the COVID-19 pandemic has not only impacted children’s and adolescents’ physical health but also their mental health and well-being. An Italian study of children and adolescents aged 6–18 years found that 61.3% of participants were more concerned about the future, 46% reported sleep difficulties, and 72.8% reported experiencing more attention difficulty than before the pandemic [6]. Another study in Germany showed that children and adolescents felt sadder and less well during this period than children and adolescents before the pandemic [7]. Their findings also indicated that well-being and sadness were positively associated with physical activities. Similar results were also found in a Canadian study on children aged 9–15 that 37.6% of participants felt bored, and 31% of them were more worried than before the pandemic in 2020 [8].

A key indicator of adolescent health development is physical activity—it affects lifelong health, including physical health (such as obesity and physical strength) and mental health (such as anxiety and depression) [9,10]. Previous studies have shown that the pandemic reduced the time spent engaging in physical activity among adolescents due to school closures and home confinement [11,12,13,14]. In addition, due to prolonged lockdowns causing higher levels of stress, children and adolescents experienced depression, anxiety, and fearful emotions, which increased the uptake of mental health counseling [15]. Physical activity is correlated with psychological health and can improve mental health [16]. According to a scoping review of adult and child studies on changes in physical activity during the pandemic, changes in physical activity levels were the most evaluated results, followed by a relation between physical and mental health issues [14]. However, studies investigating the factors that affect the positive or negative changes in physical activity and depression caused by COVID-19 are insufficient. More attention should be paid to adolescent activities and depression.

Therefore, the aims of this study are as follows:To look at the changes in physical activity and depression in adolescents due to the pandemic.To investigate characteristics related to changes in physical activity and depression among the changes in health behaviors.

## 2. Materials and Methods

### 2.1. Study Design

This study employed a descriptive correlation methodology, using raw data from the 17th Korea Youth Risk Behavior Web-based Survey (KYRBS), which was conducted in 2021.

### 2.2. Participants

The KYRBS is a nationally representative survey and is conducted annually to understand the current status of the health behaviors of adolescents. The questionnaires of the survey are focused on health status and health behaviors of adolescents and include potential health risks issues such as smoking, drinking, physical activity, eating habits, obesity, mental health, sexual behavior, internet and mobile use, drugs, and mental health

The data of 54,835 participants were used in the study by excluding 13 participants who did not respond to items concerning changes in physical activity due to COVID-19. Data were collected using a unique anonymous participant number and anonymity and confidentiality were maintained throughout data collection. Consent was obtained from all participants at the time of data collection.

### 2.3. Measures

#### 2.3.1. Changes in Health Behavior Due to COVID-19

Changes in health behavior due to COVID-19 were classified into five categories from greatly increased to greatly decreased when compared to the pre-pandemic period for a total of five areas (physical activity, having breakfast, drinking, smoking, and depression). These were recategorized as “greatly increased” and “increased” in the “increased group”, “decreased” and “greatly decreased” in the “decreased group”, and “no change” in the “no change group”.

#### 2.3.2. Demographic and Sociological Characteristics

The demographic characteristics of the subjects were analyzed based on gender, school level, residence area, changes in economic status due to COVID-19, whether participants were living with family, and subjective health status. Gender was categorized as “male” and “female”, school level was “middle school” and “high school”, and residence area was categorized as “large cities (metropolitan cities with a population of more than 500,000)” and “small and medium-sized cities or below with a total population of less than 500,000 or around 100,000”. Changes in economic status were categorized as “difficulty” and “no change” and whether participants were living with a family member was categorized as “living together” and “not living together”. For subjective health status, participants were asked “how do you rate your health?”. Responses of “very healthy”, “healthy”, “normally healthy” were classified as “healthy”, and “unhealthy” and “very unhealthy” were classified as “unhealthy”.

#### 2.3.3. Health-Related Behavior

Having breakfast, smoking, and drinking were used as health-related behavior variables. For having breakfast variables, the question “In the last 7 days, how many days did you eat breakfast (excluding milk or juice only)?” was used. Having breakfast more than 6 days a week was categorized as having breakfast regularly and having breakfast 5 days or less per week was classified as “not having breakfast regularly”. For currently smoking variables, the question “In the past 30 days, how many days have you smoked at least one regular cigarette?” was used. “No” was recorded as “non-smoking” and “more than 1–2 days a month” was recorded as “smoking”. For currently drinking variables, the question “How many days have you had at least one drink in the last 30 days?” was used. “No” was recorded as “not drinking” and “More than 1–2 days a month” was recorded as “drinking”. For physical activity, participants were asked “In the last seven days, how many days did you spend more than 60 min a day on physical activities where your heart rate increased, or you were out of breath?”. Responses of “5 days a week or more” was classified as “yes” and “4 days a week or less” was classified as “no”

#### 2.3.4. Mental Health

For mental health variables, stress, loneliness, despair, suicidal ideation, suicide plans, and suicide attempts were used. For stress, the question “How much stress do you normally feel?” was used, from “Not at all” to “Feel a little” to “no”, and “Feel a lot” to “Feel very much” to “yes”. For loneliness, the question “How often have you felt lonely in the past 12 months?” was used. “I felt lonely at all or almost never” was recorded as “no” and “sometimes, often, always felt lonely” was recorded as “yes”. For feelings of despair, the question “Have you ever felt sadness or despair enough to stop your daily life for 2 weeks in the last 12 months?” was used. It was categorized as “no” and “yes”. For suicidal ideation, the question, “Have you seriously thought of suicide in the past 12 months?” was used. It was categorized as “no” and “yes”. For the suicide plan, the question “Have you made a specific plan to commit suicide in the past 12 months?” was used. It was categorized as “no” and “yes”. Finally, for suicide attempts, the question “Have you attempted suicide in the past 12 months?” was used. It was categorized as “no” and “yes”.

### 2.4. Data Collection

The KYRBS has been published annually since 2005 to identify the status and trends of Korean adolescent health behaviors. The 17th KYRBS is an anonymous, self-reported, online survey of middle and high school students. The sampling process was conducted during the stages of population stratification, sampling distribution, and sampling. Accordingly, the population was divided into 117 strata, using the stratification variables of 39 regional groups and school levels (middle school, general high school, and specialized high school) across the country. The sample number of schools was distributed by applying the proportional distribution method so that the population composition ratio and sample composition ratio for each stratification variable matched for each of the 400 middle and high schools. For the sampling unit, a phylogenetic sampling method was used with the school as the primary sampling unit and a randomly selected class as the secondary sampling unit. Prior to data collection, sample schools and classes were selected, and student status was registered. The teachers in charge of survey support at the sample school were selected and trained. They explained the purpose of the survey to the participants who submitted written consent. All students in the sample class were surveyed excluding long-term absentees, differently abled students who could not participate in the survey on their own, and students with reading disabilities. The survey was conducted by completing a pre-set online survey using a school computer or mobile device (tablet, personal computer, smartphone). The survey took between 45 and 50 min.

### 2.5. Statistical Analysis

To explore the characteristics of the multiple-sample design, multiple-sample design data analysis was used by stratifying multistage probability sampling of the raw KYRBS data. IBM SPSS Statistics 27.0 program (IBM Corp., Armonk, NY, USA) was used to reflect the integrated layer (Strata), cluster variable (Cluster), weight (W), and finite population correction coefficient (FPC), which were present in the raw data and analyzed.

First, the demographic characteristics, health-related behaviors, and mental health of adolescents were analyzed by frequency (applied sample size) and percentage (applied weighted value).

Second, changes in health-related behaviors and mental health due to COVID-19 were analyzed using frequency (applied sample size) and percentage (applied weighted value).

Third, differences between physical activity and depression caused by COVID-19 according to the demographic, health-related behavioral, and mental health characteristics of adolescents were analyzed using a composite sample Rao–Scott χ2-test.

Fourth, to analyze the effect of changes in physical activity due to COVID-19, multiple logistic regression analysis was conducted on the group without changes in physical activity and depression as the reference group, and the increased and decreased groups were the comparison group.

## 3. Results

### 3.1. Participant Characteristics

The KYRBS research subjects were Korean middle and high school students and a total of 800 schools—400 middle schools and 400 high schools—were enrolled. The total number of subjects recruited was 59,066, from 796 schools (399 middle schools, 397 high schools), and 54,848 students participated in the survey. The teachers in charge of survey support had a large workload, and there was reduced access to the computer room due to COVID-19; therefore, the participation rate based on the number of students was 92.9%.

Table 1 shows the demographic characteristics, health-related behaviors, and mental health characteristics of participants. The gender distribution of the participants was 51.7% male and 48.3% female. At the school level, 51.0% were in middle school, 49.0% were in high school, 58.1% were in small and medium cities, and 41.9% were in large (metropolitan) cities. Regarding changes in economic conditions due to COVID-19, 70.1% of respondents said there was no change, and 29.9% said that their family economics were more difficult than before. Of the participants, 96.2% lived with their family members, while 3.8% did not live with their family or parents. Regarding subjective health status, 90.8% of participants saw themselves as healthy, while 9.2% felt unhealthy. As for currently physical activity, 14.6% of respondents were spending more than 5 days a week on physical activities, while 85.4% were spending less than that.

As for the health-related behavioral characteristics of participants, 64.8% of participants had breakfast less than five days a week, 4.5% of the participants were currently smoking, and 10.8% were currently drinking alcohol.

The results for the mental health characteristics of participants showed that 38.8% were stressed, 52.3% felt lonely, 26.8% felt hopeless, 12.7% had suicidal thoughts, 4.0% had a specific suicide plan, and 2.2% had attempted suicide.

### 3.2. Changes in Health-Related Behavior and Mental Health Due to COVID-19

Table 2 shows changes in health-related behaviors and mental health caused by COVID-19. During COVID-19, 19.4% of participants increased their physical activity while 49.1% of them had decreased physical activity. Compared to before COVID-19, the group which increased its behavior missed breakfast more by 14.3%, drank more by 2.8%, smoked more by 1%, or felt depressed more by 36.9%. Meanwhile, the group who decreased their behavior did so by missing breakfast less by 13.1%, consuming alcohol less by 15%, smoking less by 15%, or feeling depressed less by 9.7%.

### 3.3. Differences between Physical Activity and Depression Caused by COVID-19 According to the Characteristics of Adolescents

Table 3 shows the differences between physical activity and depression caused by COVID-19 according to the characteristics of adolescents. There was a statistically significant difference in physical activity in the following characteristics: gender (χ^2^ = 1667.922, *p* < 0.001), school level (χ^2^ = 368.963, *p* < 0.001), residence area (χ^2^ = 48.187, *p* < 0.001), changes in economic status due to COVID-19 (χ^2^ = 160.760, *p* < 0.001), subjective health status (χ^2^ = 402.359, *p* < 0.001), having breakfast (χ^2^ = 19.917, *p* < 0.001), current smoking (χ^2^ = 177.256, *p* < 0.001), current drinking (χ^2^ = 144.934, *p* < 0.001), stress (χ^2^ = 309.900, *p* < 0.001), loneliness (χ^2^ = 460.128 *p* < 0.001), despair (χ^2^ = 108.957, *p* < 0.001), suicidal ideation (χ^2^ = 114.371, *p* < 0.001), suicide plan (χ^2^ = 41.676, *p* < 0.001), and suicide attempts (χ^2^ = 15.851, *p* = 0.001).

Among the various characteristics of adolescents, depression showed statistically significant differences in all characteristics except residence area. In other words, there was a significant difference in gender (χ^2^ = 2135.019, *p* < 0.001), school level (χ^2^ = 292.314, *p* < 0.001), change in economic status due to COVID-19 (χ^2^ = 593.554, *p* < 0.001), living with family (χ^2^ = 66.680, *p* < 0.001), subjective health status (χ^2^ = 853.635, *p* < 0.001), having breakfast (χ^2^ = 88.202, *p* < 0.001), current smoking (χ^2^ = 24.927, *p* < 0.001), current drinking (χ^2^ = 107.126, *p* < 0.001), stress (χ^2^ = 6234.939, *p* < 0.001), loneliness (χ^2^ = 8564.969 *p* < 0.001), despair (χ^2^ = 5194.104, *p* < 0.001), suicidal ideation (χ^2^ = 3881.061, *p* < 0.001), suicide plan (χ^2^ = 941.879, *p* < 0.001), and suicide attempts (χ^2^ = 646.924, *p* < 0.001).

### 3.4. Factors Affecting Changes in Physical Activity and Depression Due to COVID-19

Before conducting logistic regression analysis, we examined the confounding variables that affect physical activity and depression due to COVID-19. These variables were gender, school of type, region of residence, change in economic status due to COVID-19, reside with family, and subjective health status. After controlling these variables and using “no change” in physical activity and depression as the reference group, the factors affecting changes in physical activity and depression due to COVID-19 are shown in Table 4.

The number of adolescents with increased physical activity compared to before COVID-19 were lower in having breakfast (aOR = 0.93), currently drinking (aOR = 0.85), loneliness (aOR = 0.90), despair (aOR = 0.82), and suicidal plan (aOR = 0.76) than those who reported no change. Compared to the pre-pandemic, adolescents whose physical activity decreased were higher in current smoking (aOR = 1.53), current drinking (aOR = 1.15), stressed (aOR = 1.24), and attempted suicide (aOR = 1.33) than those who reported no change. On the other hand, adolescents whose physical activity decreased were lower in having breakfast (aOR = 0.90), loneliness (aOR = 0.70), and suicidal ideation (aOR = 0.83).

Compared to the pre-pandemic, adolescents with increased rates of depression were higher among students who were currently smoking (aOR = 1.31) and experiencing stress (aOR = 2.38). On the other hand, loneliness (aOR = 0.27), despair (aOR = 0.56), suicidal ideation (aOR = 0.49), and attempted suicide (aOR = 0.84) had lower indications of depression. Compared to the pre-pandemic period, the group with reduced depression was higher among those having regular breakfast (aOR = 1.19), stressed adolescents (aOR = 1.12), and those who experienced loneliness (aOR = 1.09). On the other hand, it was found to be lower among students in feeling despair (aOR= 0.90) and experiencing suicide plans (aOR = 0.73).

## 4. Discussion

This study used raw data from the 17th KYRBS to identify changes in adolescents’ health behaviors due to COVID-19 and analyzed the characteristics related to changes in physical activity and depression among these changes in health behaviors. It was found that drinking and smoking among adolescents before and after the pandemic showed a positive change, with the decreased group (15% each) being higher than the increased group (drinking 2.8%, smoking 1%). Not having breakfast remained at similar levels for both the decreased group (13.1%) and increased group (14.3%), but physical activity was higher in the decreased group (49.1%) than in the increased group (19.4%), and the rate of experiencing depression was higher in the increased group (36.9%) than in the decreased group (9.7%). This indicates that COVID-19 is associated with a decrease in physical activity and an increase in depression among adolescents. The decrease in drinking and smoking in adolescents is consistent with the results of the National Survey on Drug Use and Health (NSDUH) conducted among US adolescents (12–17 years old), which shows the current smoking rate (from 2.3% in 2019 to 1.4% in 2020) and the current drinking rate (from 9.4% in 2019 to 8.2% in 2020) were decreased [17].

Many previous studies have reported an increase in sedentary activity time and a decrease in physical activity time [11,12,13,14,18,19] and various psychological problems, including depression and anxiety during the pandemic [19,20]. These changes in adolescent health behavior were related to social distancing according to the rules during the pandemic, namely suspension of daily educational activities, such as postponement of school openings, remote classes, home confinement, and reduction of interaction outside the home [16,18,21]. In the case of adolescents, school life and interaction with peers are important factors that positively affect various social, physiological, and psychological adaptations [22]. A study of adolescents in southern China conducted immediately after schools reopened found that those with higher levels of physical activity were less likely to show symptoms of depression, compared to those with lower levels of physical activity [23]. Although the WHO recommends vigorous physical activity of about 60 min a day in adolescence, 81% of adolescents aged 11 to 17 worldwide lack physical activity [24]. In the case of Korean adolescents, only 13.9% of adolescents participated in physical activity for at least 60 min a day 5 days a week, which was among the lowest in the world before the pandemic [25,26]. As this lack of physical activity has worsened during the pandemic [22], it is necessary to prepare various ways to promote youth physical activity post-pandemic.

In this study, the result shows that the group with changes in physical activity due to COVID-19 was 68.5% which is more than twice as high as the group without change (31.5%), and the group with reduced physical activity was more than the increased group. The group with depression changes due to the pandemic was 46.6% which is lower than the group without change (53.4%), and the group with increased depression was higher than the reduced group. In addition, the factors affecting the increasing and decreasing groups of physical activity and depression were different. Among health-related behaviors, having regular breakfast was associated with a decrease in depression; current smoking was associated with a decrease in physical activity but an increase in depression; and current drinking was associated with a decrease in physical activity. Among the mental health characteristics, it was confirmed that stress was associated with a decrease in physical activity and an increase in depression. In addition, it was found that loneliness was associated with a decrease in depression while suicide attempt was associated with a decrease in physical activity. Considering these variables, it is important to find a way to improve the physical and mental health of adolescents during the pandemic period.

Compared to the group with no change in having breakfast, the group with increased physical activity had breakfast 0.93 times less, the group with reduced physical activity had it 0.90 times less, and the group with reduced depression had it 1.19 times more. Current smoking was 1.53 times higher in the decreased physical activity group and 1.31 times higher in the increased depression group. Current drinking was 0.85 times lower in the increased physical activity group and 1.15 times higher in the decreased physical activity group.

This differs from a study [21] that found irregular breakfast as a factor related to obesity and depression in adolescents during the pandemic and current smoking and current drinking as factors related to the experience of depression. School meals were stopped when school was suspended due to the pandemic, and local care services weakened, and this resulted in children missing meals at an increased rate, and they experienced a care gap [27,28] and the obesity level of children and adolescents worsened. When indices of metabolic syndromes, such as cholesterol and triglycerides, are elevated [4] it may affect physical health; therefore, it is necessary to manage the diet of adolescents who are at risk of under-nutrition. In addition, looking at the modifiable risk and protective factors of adolescents identified in previous studies, physical activity and a healthy diet reduced depression [29], having a daily breakfast was significantly associated with better mental health outcomes [30], and high alcohol consumption and smoking were associated with higher levels of depression [29,30,31,32]. To reduce the deterioration of dietary life indicators, the increase in obesity rate, and depression in adolescents before and after the pandemic, it is necessary to continuously evaluate the impact of lifestyle changes related to the pandemic and create an educational environment to promote health of youth.

Compared to the group with no change in stress, the reduced physical activity group was 1.24 times more stressed, the increased depression group was 2.38 times more stressed, and the reduced depression group was 1.12 times more stressed among all groups. This is similar to a study [32] in which participants who reported negative changes in physical activity had higher levels of stress symptoms. Likewise, it was similar to another study [21], where stress perception was found to be a factor related to the experience of depression. In addition, pandemic-related stress has been reported to have a significant effect on physical activity levels and depression [25].

Compared to the group with no change in the pandemic, the number of adolescents who experienced loneliness was 0.90 times lower in the increased physical activity group, 0.70 times lower in the decreased physical activity group, 0.27 times lower in the increased depression group, and 1.09 times higher in the decreased depression group. Compared to the group with no change, the number of adolescents who experienced despair was 0.82 times lower in the increased physical activity group, 0.56 times lower in the increased depression group, and 0.90 times lower in the depression decreased group. In previous studies, social isolation was found to increase loneliness in some people, and it was associated with increased depression and suicidal thoughts [33,34]. In addition, the duration of loneliness showed a stronger correlation with mental health symptoms than the intensity of loneliness [34]. In particular, the severance of interpersonal relationships has a great influence on psychological changes in adolescents. A decrease in social interaction is a major risk factor for mental health, and disconnection from social relationships, that is, loneliness, negatively affects the physical and mental health of adolescents [35]. Therefore, it is necessary to provide preventive support and early intervention, where possible, during social isolation, including forced isolation, and mitigation of mental health problems.

The frequency of suicidal ideation, suicide plans, and suicide attempts were significantly higher among the decreased physical activity and increased depression groups than in the other groups. However, as shown in the multiple logistic regression analysis, the number of adolescents having suicidal thoughts was 0.83 times lower in the decreased physical activity group and 0.49 times lower in the increased depression group, respectively, compared to the group with no change before and after COVID-19. The number of adolescents with experience of suicide planning were 0.76 times lower in the physical activity increased group and 0.73 times lower in the depression decreased group, respectively. The number of adolescents who had attempted suicide were found to be 1.33 times higher in the decreased physical activity group, confirming an association between suicide attempts and reduced physical activity. In a meta-analysis of suicidal behavior during the pandemic [36], when compared with the event rates in pre-pandemic studies, suicidal ideation (4.68%), suicide attempts (4.68%), and self-harm (9.63%) increased during the pandemic, and it differed according to age (younger people), gender (women), and geopolitics (individuals from democratic countries). However, in a study investigating the association between the pandemic and suicide-related behavior among Korean adolescents, there were fewer suicide-related behaviors, including suicidal thoughts, suicide plans, and suicide attempts, in the 2020 study than in the 2019 study [37]. Previous studies have reported a decrease in suicides since the start of the pandemic [38], and rates of suicidal ideation and attempts in pediatric emergency departments were higher for several months in 2020 compared to 2019, but this was due to COVID-19-related stress, which coincided with the period when the community response was high [39]. As such, the rate of suicide-related behaviors may vary depending on the stage of the pandemic. Therefore, it is necessary to identify the trend of suicidal behavior during the pandemic, the long-term mental health, and economic impact of the pandemic, and increase preparedness to respond to changes in the situation. In addition, there are many studies examining the relationship between depression and suicide-related behaviors, but they do not pay attention to the relationship between changes in physical activity levels and suicide attempts. Therefore, it is necessary to investigate the relationship between physical activity and suicide-related behaviors.

Demographic variables that showed statistical differences in physical activity and depression could act as confounding variables, so they were controlled for multiple logistic regression analysis. Among these variables, we found that changes in economic status due to COVID-19 and subjective health status affect changes in physical activity and depression in adolescents and show a similar pattern of increase or decrease. However, it was confirmed that the increase or decrease in physical activity or depression showed distinctly different patterns according to gender and school level. In other words, the group that showed an increase in physical activity and a decrease in depression had a high proportion of male and middle school students, while the opposite patterns were found in female and high school students.

Studies conducted during the pandemic have analyzed the relationship between physical activity and gender [11,40]. A study that analyzed the level and characteristics of moderate-to-vigorous physical activity (MVPA), according to gender and number of school physical activities for middle school students during the pandemic found that female students’ physical activity levels were lower than those of male students [22]. Although it was relatively low, it has been reported that the decrease in physical activity in men was greater than that in women. International studies that longitudinally analyzed the level of physical activity before and during the pandemic also reported that the decrease in physical activity in men during the pandemic was greater than that in women [11,40].

Research has suggested that female adolescents perform light exercise, such as home training, to control their increased weight during the pandemic [41]. In COVID-19, it has been reported that stress is higher among those who perform a high level of low-intensity physical activity [25]; therefore, it is necessary to plan a physical activity program suitable for these activity differences (intensity and duration) according to gender. In addition, previous studies have shown that male students have higher levels of loneliness, anxiety, and depression due to COVID-19 than female students [25], which is in contrast with the results of this study. There is a suggestion that male students who participate in competitive activities suffer from negative psychological states due to the restrictions on physical activities [42]. However, for female students, who have lower levels of stress, loneliness, anxiety, and depression, there is a possibility that they are satisfied with light and enjoyable sports or activities during remote learning. Thus, to increase positive psychological factors, it is necessary to consider gender and the level of physical activity.

A previous study found that 17-year-old high school students had significantly lower levels of physical activity than 13-year-old or 15-year-old middle school students [43]. However, Choi et al. showed that the change in physical activity practice rate between the first and second years of the pandemic increased again for middle school students (1.5%p) and among female students increased only in middle school students (0.7%p). In the case of the depression rate during the pandemic, it decreased by (–2.2%p, –1.0%p) for female middle and high school students but increased among male and female middle school students (3.9%p, 2.0%p) [21]. In addition to the school level, the timing of the survey and the gender of the participants have an impact on results; thus, it is necessary to consider gender, school level, and the timing of the survey in future research.

The strength of this study is that it analyzes data on changes in physical activity and depression in adolescents and the factors that are associated with those changes. In particular, this study shows a difference in the factors that may associate with the increase and decrease of physical activity and depression. This needs to be considered when planning a program to promote physical activity and reduce depression among adolescents after the pandemic. As for the limitation of this study, it used secondary data and subjective indicators in relation to the changes in health behaviors rather than objective indicators of physical activity or depression. Therefore, this must be taken into consideration when interpreting the results. In addition, there is a limit to generalizing to other Asian adolescents as country-specific response guidelines in the pandemic situation may have different results on adolescents’ physical activity or depression.

## 5. Conclusions

This study observed positive changes in drinking and smoking among adolescents before and after the pandemic, with a larger number of adolescents in the decreased group rather than in the increased group. However, there were many negative changes related to physical activity and depression in the decreased and increased groups, respectively. The changes in physical activity and depression, which showed negative changes due to COVID-19, were related to complex factors such as health-related behaviors and mental health, and it was found that the related factors differed depending on whether the group behavior increased or decreased. As revealed in the results of this study, it is necessary to develop an educational environment and program for youth health promotion by considering factors affecting physical activity and depression. It is also necessary to develop strategies and programs for promotion of physical activity in anticipation of future disruptions resulting from the pandemic.

## Figures and Tables

**Table 1 healthcare-11-00517-t001:** Participants’ characteristics (*n* = 54,835).

Characteristics	Categories	*n* (%)
Gender	Male	28,393 (51.7)
	Female	26,442 (48.3)
School type	Middle school	30,006 (51.0)
	High school	24,829 (49.0)
Region of residence	M-S and rural city	30,978 (58.1)
	Metropolitan	23,857 (41.9)
Economic change with COVID-19	None	38,128 (70.1)
	Yes	16,707 (29.9)
Reside with family	No	2421 (3.8)
	Yes	52,414 (96.2)
Subjective health status	Unhealthy	5020 (9.2)
	Healthy	49,815 (90.8)
Current physical activity	None	46,486 (85.4)
	Yes	8349 (14.6)
Eat breakfast	No	35,649 (64.8)
	Yes	19,186 (35.2)
Current Smoking	None	52,428 (95.5)
	Yes	2407 (4.5)
Current drinking	None	49,012 (89.2)
	Yes	5822 (10.8)
Stress	None	33,593 (61.2)
	Yes	21,242 (38.8)
Loneliness	None	26,315 (47.7)
	Yes	28,520 (52.3)
Despair	None	40,148 (73.2)
	Yes	14,687 (26.8)
Suicidal ideation	None	47,880 (87.3)
	Yes	6955 (12.7)
Suicidal plan	None	52,630 (96.0)
	Yes	2205 (4.0)
Suicidal attempt	None	53,590 (97.8)
	Yes	1245 (2.2)

M-S = medium- and small-sized; *n* (%) = *n*: unweighted; %: weighted.

**Table 2 healthcare-11-00517-t002:** Changes in health-related behavior and mental health due to COVID-19 (*n* = 54,835).

Variables	Categories	*n* (%)
Physical activity	Increased change	11,094 (19.4)
	No change	17,740 (31.5)
	Decreased change	26,001 (49.1)
Missing breakfast	Increased change	7912 (14.3)
	No change	39,791 (72.6)
	Decreased change	7132 (13.1)
Drinking	Increased change	1518 (2.8)
	No change	44,659 (82.1)
	Decreased change	8425 (15.0)
Smoking	Increased change	527 (1.0)
	No change	45,078 (83.9)
	Decreased change	8330 (15.0)
Depression	Increased change	19,730 (36.9)
	No change	29,562 (53.4)
	Decreased change	5543 (9.7)

*n* (%) = *n*: unweighted; %: weighted.

**Table 3 healthcare-11-00517-t003:** Participants’ characteristics, health-related behaviors, mental health, and relationship to COVID-19 physical activity change and depression change (*n* = 54,835).

Variables	Categories	Physical Activity				Depression			
		No Change (*n* = 17,740) *n* (%)	Increase Group (*n* = 11,094) *n* (%)	Decrease Group (*n* = 26,001) *n* (%)	χ^2^ (*p*)	No Change (*n* = 29,562) *n* (%)	Increase Group (*n* = 19,730) *n* (%)	Decrease Group (*n* = 5543) *n* (%)	χ^2^ (*p*)
Gender	Male	9375 (53.3)	7492 (67.6)	11,526 (44.4)	1667.922 (<0.001)	16,962 (57.5)	7679 (39.3)	3752 (66.9)	2135.019 (<0.001)
	Female	8365 (46.7)	3602 (32.4)	14,475 (55.6)		12,051 (42.5)	12,051 (60.7)	1791 (33.1)	
School type	Middle school	9355 (48.3)	7030 (59.3)	13,621 (49.4)	368.963 (<0.001)	16,169 (50.8)	10,221 (48.4)	3616 (61.6)	292.314 (<0.001)
	High school	8385 (51.7)	4064 (40.7)	12,380 (50.6)		13,393 (49.2)	9509 (51.6)	1927 (38.4)	
Region of residence	M-S & rural city	10,303 (59.3)	6515 (59.8)	14,160 (56.6)	48.187 (<0.001)	16,669 (57.8)	11,105 (58.2)	3204 (59.1)	3.203 (0.843)
	Metropolitan	7437 (40.7)	4579 (40.2)	11,841 (43.4)		12,893 (42.2)	8625 (41.8)	2339 (40.9)	
Economic change with COVID-19	None	12,970 (73.6)	7398 (67.1)	17,760 (69.1)	160.760 (<0.001)	21,867 (74.5)	12,537 (64.4)	3724 (67.7)	593.554 (<0.001)
	Yes	4770 (26.4)	3696 (32.9)	8241 (30.9)		7695 (25.5)	7193 (35.6)	1819 (32.3)	
Reside with family	No	779 (3.8)	532 (4.4)	1110 (3.6)	13.999 (0.053)	28,440 (96.7)	18,663 (95.3)	5311 (96.6)	66.680 (<0.001)
	Yes	16,961 (96.2)	10,562 (95.6)	24,891 (96.4)		1122 (3.3)	1067 (4.7)	232 (3.4)	
Subjective health status	Unhealthy	1365 (7.6)	637 (5.7)	3018 (11.7)	402.359 (<0.001)	27,696 (93.6)	16,953 (86.0)	5166 (92.9)	853.635 (<0.001)
	Healthy	16,375 (92.4)	10,457 (94.3)	22,983 (88.3)		1866 (6.4)	2777 (14.0)	377 (7.1)	
Eat breakfast	No	11,734 (66.0)	7235 (65.1)	16,680 (63.9)	19.917 (<0.001)	18,711 (63.0)	13,231 (66.8)	3707 (67.3)	88.202 (<0.001)
	Yes	6006 (34.0)	3859 (34.9)	9321 (36.1)		10,851 (37.0)	6499 (33.2)	1836 (32.7)	
Current Smoking	None	16,872 (94.9)	10,443 (93.7)	25,113 (96.6)	177.256 (<0.001)	28,393 (95.9)	18,741 (95.0)	5294 (95.4)	24.927 (<0.001)
	Yes	868 (5.1)	651 (6.3)	888 (3.4)		1169 (4.1)	989 (5.0)	249 (4.6)	
Current drinking	None	15,820 (88.9)	9626 (86.4)	23,566 (90.6)	144.934 (<0.001)	26,769 (90.4)	17,268 (87.5)	4975 (89.5)	107.126 (<0.001)
	Yes	1920 (11.1)	1468 (13.6)	2434 (9.4)		2793 (9.6)	2461 (12.5)	568 (10.5)	
Stress	None	11,595 (65.4)	7039 (63.7)	14,959 (57.6)	309.900 (<0.001)	7669 (25.7)	11,974 (60.2)	1599 (28.9)	6234.939 (<0.001)
	Yes	6145 (34.6)	4055 (36.3)	11,042 (42.4)		21,893 (74.3)	7756 (39.8)	3944 (71.1)	
Loneliness	None	9539 (53.6)	5526 (49.2)	11,250 (43.3)	460.128 (<0.001)	18,557 (62.8)	4222 (21.9)	3536 (62.7)	8564.969 (<0.001)
	Yes	8201 (46.4)	5568 (50.8)	14,751 (56.7)		11,005 (38.0)	15,508 (78.1)	2007 (37.3)	
Despair	None	13,527 (76.0)	7878 (71.0)	18,743 (72.3)	108.957 (<0.001)	24,826 (84.0)	10,806 (55.4)	4516 (81.3)	5194.104 (<0.001)
	Yes	4213 (24.0)	3216 (29.0)	7258 (27.7)		4736 (16.0)	8924 (44.6)	1027 (18.7)	
Suicidal ideation	None	15,865 (89.4)	9683 (87.2)	22,332 (86.0)	114.371 (<0.001)	27,871 (94.4)	14,858 (75.7)	5151 (92.3)	3881.061 (<0.001)
	Yes	1875 (10.6)	1411 (12.8)	3669 (14.0)		1691 (5.6)	4872 (24.3)	392 (7.7)	
Suicidal plan	None	17,145 (96.6)	10,561 (95.0)	24,924 (95.9)	41.676 (<0.001)	29,016 (98.1)	18,242 (92.6)	5372 (96.8)	941.879 (<0.001)
	Yes	595 (3.4)	533 (5.0)	1077 (4.1)		546 (1.9)	1488 (7.4)	171 (3.2)	
Suicidal attempt	None	17,367 (97.9)	10,796 (97.3)	25,427 (97.9)	15.851 (0.001)	29,281 (99.1)	18,853 (95.7)	5456 (98.4)	646.924 (<0.001)
	Yes	373 (2.1)	298 (2.7)	574 (2.1)		281 (0.1)	877 (4.3)	87 (1.6)	

M-S = medium- and small sized; *n* (%) = *n*: unweighted; %: weighted.

**Table 4 healthcare-11-00517-t004:** Multinomial logistic regression for predictors of COVID-19 exchange physical activity and depression among adolescents (*n* = 54,835).

Variables	Physical Activity				Depression			
	Increase Group (*n* = 11,094)		Decrease Group (*n* = 26,001)		Increase Group (*n* = 19,730)		Decrease Group (*n* = 5543)	
	aOR (95% CI)	*p*	aOR (95% CI)	*p*	aOR (95% CI)	*p*	aOR (95% CI)	*p*
Eat breakfast	0.93 (0.88, 0.98)	0.005	0.90 (0.86, 0.93)	<0.001	0.99 (0.95, 1.03)	0.645	1.19 (1.11, 1.27)	<0.001
Current smoking	0.93 (0.83, 1.05)	0.265	1.53 (1.38, 1.71)	<0.001	1.31 (1.18, 1.46)	<0.001	0.98 (0.83, 1.17)	0.833
Current drinking	0.85 (0.78, 0.92)	<0.001	1.15 (1.07, 1.24)	<0.001	1.01 (0.94, 1.08)	0.837	0.95 (0.86, 1.06)	0.394
Stress	0.95 (0.90, 1.01)	0.093	1.24 (1.18, 1.30)	<0.001	2.38 (2.28, 2.49)	<0.001	1.12 (1.04, 1.20)	0.003
Loneliness	0.90 (0.85, 0.95)	<0.001	0.70 (0.67, 0.73)	<0.001	0.27 (0.25, 0.28)	<0.001	1.09 (1.02, 1.16)	0.013
Despair	0.82 (0.77, 0.88)	<0.001	1.03 (0.98, 1.09)	0.246	0.56 (0.53, 0.58)	<0.001	0.90 (0.83, 0.97)	0.009
Suicidal ideation	0.99 (0.90, 1.09)	0.870	0.83 (0.76, 0.89)	<0.001	0.49 (0.45, 0.53)	<0.001	0.88 (0.77, 1.01)	0.064
Suicidal plan	0.76 (0.66, 0.88)	0.001	1.00 (0.89, 1.14)	0.948	1.04 (0.90, 1.20)	0.610	0.73 (0.59, 0.90)	0.003
Suicidal attempt	1.09 (0.90, 1.32)	0.369	1.33 (1.19, 1.62)	<0.001	0.84 (0.71, 0.99)	0.042	0.82 (0.63, 1.08)	0.156

Notes. The reference group was adolescents who reported no change; aOR = adjusted odds ratio, CI = 95% confidence interval.

## Data Availability

No new data were created or analyzed in this study. Data sharing is not applicable to this article.

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
