# Peer review of "Changes in Physical Activity and Depression among Korean Adolescents Due to COVID-19: Using Data from the 17th (2021) Korea Youth Risk Behavior Survey"

_healthcare, 2023, doi:10.3390/healthcare11040517_

Round 1

Reviewer 1 Report

I would like to thak the editor for the opportunity to review this work. 

Introduction 

I believe that the introduction is clear and well written, i suggest the authors to include more references on the effect of COVID-19 in children and adolescent also considering Europe situation like : 

Salussolia A, Lenzi J, Montalti M, et al. Physical Well-Being of Children and Adolescents during the SARS-CoV-2 Pandemic: Findings from the "Come te la Passi?" Cross Sectional Survey in Bologna, Italy. Children (Basel). 2022;9(12):1950. Published 2022 Dec 12. doi:10.3390/children9121950

Dallolio, Laura et al. “The impact of COVID-19 on physical activity behaviour in Italian primary school children: a comparison before and during pandemic considering gender differences.” BMC public health vol. 22,1 52. 8 Jan. 2022, doi:10.1186/s12889-021-12483-0

In general, the study appears to be important in the scenario of the Covid-19 effect among children. I suggest the author to rephrase the goal of the study specifying the main and the secondary goal. 

Methods section

Lines 57-59 Author should cut information already mentioned in the introduction like the study aim.

2.2 Particpants 

Please, moved the data regarding the description of the sample in the first part of the result section. Here authors should explain how they performed the enrollment with no specific number that should be include in the results section, as the STROBE (guidelins for oservational study) wants.

2.3.2. Demographic and sociological characteristics

I suggest that authors include the explanaton on what they considered metropolitan area (i.e., number of residential peole) How author decided what is an health status? This variables is self repored? The questionnaire should be briefly describe in order to allow the reader understanding the data assessment.

Results and Discussion 

Results are clear reported, i suggest the author to improve the tables layout.

Author Response

Dear Reviewer

I sincerely appreciate all your valuable comments and suggestions. In the table below, our responses to the Reviewers’ comments are described in a point-to-point manner. Appropriated changes suggested by the Reviewers have been introduced to the manuscript.

We look forward to your reply.

Sincerely,

Reviewer 1

Comments

Responses

1.

I suggest the authors to include more references on the effect of COVID-19 in children and adolescent also considering Europe situation like: Salussolia A, Lenzi J, Montalti M, et al. Physical Well-Being of Children and Adolescents during the SARS-CoV-2 Pandemic: Findings from the "Come te la Passi?" Cross Sectional Survey in Bologna, Italy. Children (Basel). 2022;9(12):1950. Published 2022 Dec 12. doi:10.3390/children9121950

Dallolio, Laura et al. “The impact of COVID-19 on physical activity behaviour in Italian primary school children: a comparison before and during pandemic considering gender differences.” BMC public health vol. 22,1 52. 8 Jan. 2022, doi:10.1186/s12889-021-12483-0

Thank you for your comment. More references have been introduced in the introduction including suggested references: Italian study (Salussolia et al., 2022), German study (Braksie et al., 2022), and Canadian study (Mitra et al., 2021). We also added Dallolio et al., 2022 for gender difference.

(lines 36-46)

2.

In general, the study appears to be important in the scenario of the Covid-19 effect among children. I suggest the author to rephrase the goal of the study specifying the main and the secondary goal.

Thank you for your comment. We rephrased the main goal of the study and the secondary goal.

 (lines61-65)

3.

Lines 57-59 Author should cut information already mentioned in the introduction like the study aim.

Revised.

(lines 71)

4.

2.2 participants:

Please, moved the data regarding the description of the sample in the first part of the result section. Here authors should explain how they performed the enrolment with no specific number that should be include in the results section, as the STROBE (guidelines for observational study) wants.

Thank you for pointing this. We added how they performed the enrolment and moved the data regarding the description of the sample in the first part of the result section.

(lines 73-77), (lines 182-187)

5.

2.3.2. Demographic and sociological characteristics

I suggest that authors include the explanation on what they considered metropolitan area (i.e., number of residential people) How author decided what is a health status? This variable is self-reported. The questionnaire should be briefly described in order to allow the reader understanding the data assessment.

Thank you for your comment. We defined the types of residence. Health status here means self-reported health. Added clear explanation in the text.

(lines 98-99, 102-103)

6.

Results and Discussion

Results are clear reported; I suggest the author to improve the tables layout.

Table layout has been modified.

Reviewer 2

Comments

Responses

1.

The methods are presented but are incomplete (see specific comments below).

Specific comments: You describe the questions used to generate estimates for the health behaviours of breakfast, smoking, and drinking, but you do not list the questions used to generate the physical activity variables. The KYRBS surveys are patterned after the US YRBSS -- so, please report the actual physical activity patterns for your sample first, and then proceed to describe how you categorized into unchanged, decreased, increased.

Thank you for your comment. We defined the physical activity variables and described the actual physical activity patterns in Table 1.

(lines 118-121)

2

Data analysis using the Chi Square statistic is fine, but there needs to be identification which version of Chi Square (e.g., Yates, Mantel-Haenszel) is being used

Thank you for your comment. We identified the version of Chi Square (Rao-Scott).

(lines 173)

3.

The use of logistic regression is appropriate for these categorical data, but the expression of the findings needs improvement.

Thank you for your comment. We improved the expression of the findings.

(lines 242-264)

4.

The discussion is rather lengthy and tends to stray from the direct findings from the study and supplement with findings from elsewhere in the literature. Where possible, please highlight your findings first, especially the physical activity findings, and then support your interpretation of these findings with citations from the existing literature

Thank you for your comment. We revised to highlight your findings at lines 303 ~317.

5.

The synergism between depressive signs/symptoms and levels of physical activity need to be discussed in light of a dose response- where measured physical activity - whether self-report or objective, should be discussed beyond the stratification by decrease, no change, or increase

Thank you for your comment. We explained the synergism between depressive signs/symptoms and levels of physical activity at lines 291-295.

6.

Specific comments:You describe the questions used to generate estimates for the health behaviors of breakfast, smoking, and drinking, but you do not list the questions used to generate the physical activity variables. The KYRBS surveys are patterned after the US YRBSS -- so, please report the actual physical activity patterns for your sample first, and then proceed to describe how you categorized into unchanged, decreased, increased.

Thank you for your comment. We defined the physical activity variables and described the actual physical activity patterns in Table 1.

(lines 118-121)

7.

Also, using logistic regression you can look within each subgroup (e.g., males, females, etc.) and report out what, if any changes, are significant within these groups during the pandemic controlling for any types of confounders. Confounders in your data set no doubt include geographic location, SES/Income/, type of school, etc. These all need to be controlled for in your analyses.

The analytic approach needs to include the calculation of the Odds Ratios (OR) and the 95% Confidence Intervals (CI). This provides insight into the precision of your estimates and not just probabilities. This is a much better approach to analyzing categorical data. The use of logistic regression is appropriate but needs to be supported by identifying very specifically which co-variates you have controlled for in your analyses. The use of a stratified univariate approach without controlling for potential confounders is not sufficient with such data

Thank you for your comment. We controlled confounders in our data. The results of this analysis are presented in Table 4.

8.

Some discussion about the generalizability of your data to youth in South Korea or other Asian youth needs to be mentioned. Strength or limitation?

Thank for your comment. We added discussion about the generalization of our data at line 445-447.

9.

This is a very important study and has the potential of informing the healthcare and public health communities about the negative impact of most countries' response to the COVID-19 pandemic on youth and essential information going forward addressing the next pandemic with greater success and less physical and mental harm.

Thank for your comment. We believe that this study could inform the healthcare and local health communities about maintaining physical activity and decreasing sitting time is crucial in preventing mood disorders such as depression and anxiety in future epidemics that may come.

Reviewer 2 Report

General Comments:  The current manuscript seeks to examine self-reported changes in physical activity and depressive signs during the COVID-19 pandemic and their potential correlates among Korean youth in middle and high school, through the online Korean Youth Risk Behavior Surveys, collected in 2021.  The authors provide sufficient background information with appropriate citations describing the public health importance of such a study.  The methods are presented but are incomplete (see specific comments below).  Data analysis using the Chi Square statistic is fine, but there needs to be identification which version of Chi Square (e.g., Yates, Mantel-Haenszel) is being used.  The use of logistic regression is appropriate for these categorical data, but the expression of the findings needs improvement.  The discussion is rather lengthy and tends to stray from the direct findings from the study and supplement with findings from elsewhere in the literature.  Where possible, please highlight your findings first, especially the physical activity findings, and then support your interpretation of these findings with citations from the existing literature.  The synergism between depressive signs/symptoms and levels of physical activity need to be discussed in light of a dose response- where measured physical activity - whether self-report or objective, should be discussed beyond the stratification by decrease, no change, or increase.  

Specific comments:

You describe the questions used to generate estimates for the health behaviors of breakfast, smoking, and drinking, but you do not list the questions used to generate the physical activity variables.  The KYRBS surveys are patterned after the US YRBSS -- so, please report the actual physical activity patterns for your sample first, and then proceed to describe how you categorized into unchanged, decreased, increased.  Also, using logistic regression you can look within each subgroup (e.g., males, females, etc.) and report out what, if any changes, are significant within these groups during the pandemic controlling for any types of confounders.  Confounders in your data set no doubt include geographic location, SES/Income/, type of school, etc.  These all need to be controlled for in your analyses. 

The analytic approach needs to include the calculation of the Odds Ratios (OR) and the 95% Confidence Intervals (CI).  This provides insight into the precision of your estimates and not just probabilities.  This is a much better approach to analyzing categorical data.  The use of logistic regression is appropriate but needs to be supported by identifying very specifically which co-variates you have controlled for in your analyses.  The use of a stratified univariate approach without controlling for potential confounders is not sufficient with such data.  

Some discussion about the generalizability of your data to youth in South Korea or other Asian youth needs to be mentioned.  Strength or limitation?

This is a very important study and has the potential of informing the healthcare and public health communities about the negative impact of most countries' response to the COVID-19 pandemic on youth and essential information going forward addressing the next pandemic with greater success and less physical and mental harm.

Author Response

Dear Reviewer

I sincerely appreciate all your valuable comments and suggestions. In the table below, our responses to the Reviewers’ comments are described in a point-to-point manner. Appropriated changes suggested by the Reviewers have been introduced to the manuscript.

We look forward to your reply.

Sincerely,

Reviewer 2

Comments

Responses

1.

The methods are presented but are incomplete (see specific comments below).

Specific comments: You describe the questions used to generate estimates for the health behaviours of breakfast, smoking, and drinking, but you do not list the questions used to generate the physical activity variables. The KYRBS surveys are patterned after the US YRBSS -- so, please report the actual physical activity patterns for your sample first, and then proceed to describe how you categorized into unchanged, decreased, increased.

Thank you for your comment. We defined the physical activity variables and described the actual physical activity patterns in Table 1.

(lines 118-121)

2

Data analysis using the Chi Square statistic is fine, but there needs to be identification which version of Chi Square (e.g., Yates, Mantel-Haenszel) is being used

Thank you for your comment. We identified the version of Chi Square (Rao-Scott).

(lines 173)

3.

The use of logistic regression is appropriate for these categorical data, but the expression of the findings needs improvement.

Thank you for your comment. We improved the expression of the findings.

(lines 242-264)

4.

The discussion is rather lengthy and tends to stray from the direct findings from the study and supplement with findings from elsewhere in the literature. Where possible, please highlight your findings first, especially the physical activity findings, and then support your interpretation of these findings with citations from the existing literature

Thank you for your comment. We revised to highlight your findings at lines 303 ~317.

5.

The synergism between depressive signs/symptoms and levels of physical activity need to be discussed in light of a dose response- where measured physical activity - whether self-report or objective, should be discussed beyond the stratification by decrease, no change, or increase

Thank you for your comment. We explained the synergism between depressive signs/symptoms and levels of physical activity at lines 291-295.

6.

Specific comments:You describe the questions used to generate estimates for the health behaviors of breakfast, smoking, and drinking, but you do not list the questions used to generate the physical activity variables. The KYRBS surveys are patterned after the US YRBSS -- so, please report the actual physical activity patterns for your sample first, and then proceed to describe how you categorized into unchanged, decreased, increased.

Thank you for your comment. We defined the physical activity variables and described the actual physical activity patterns in Table 1.

(lines 118-121)

7.

Also, using logistic regression you can look within each subgroup (e.g., males, females, etc.) and report out what, if any changes, are significant within these groups during the pandemic controlling for any types of confounders. Confounders in your data set no doubt include geographic location, SES/Income/, type of school, etc. These all need to be controlled for in your analyses.

The analytic approach needs to include the calculation of the Odds Ratios (OR) and the 95% Confidence Intervals (CI). This provides insight into the precision of your estimates and not just probabilities. This is a much better approach to analyzing categorical data. The use of logistic regression is appropriate but needs to be supported by identifying very specifically which co-variates you have controlled for in your analyses. The use of a stratified univariate approach without controlling for potential confounders is not sufficient with such data

Thank you for your comment. We controlled confounders in our data. The results of this analysis are presented in Table 4.

8.

Some discussion about the generalizability of your data to youth in South Korea or other Asian youth needs to be mentioned. Strength or limitation?

Thank for your comment. We added discussion about the generalization of our data at line 445-447.

9.

This is a very important study and has the potential of informing the healthcare and public health communities about the negative impact of most countries' response to the COVID-19 pandemic on youth and essential information going forward addressing the next pandemic with greater success and less physical and mental harm.

Thank for your comment. We believe that this study could inform the healthcare and local health communities about maintaining physical activity and decreasing sitting time is crucial in preventing mood disorders such as depression and anxiety in future epidemics that may come.

Round 2

Reviewer 2 Report

Thank you for your revised manuscript.  The paper is much improved and the results are highlighted nicely in your tables.